# Iron, Zinc, Copper, Cadmium, Mercury, and Bone Tissue

**DOI:** 10.3390/ijerph20032197

**Published:** 2023-01-26

**Authors:** Żaneta Ciosek, Karolina Kot, Iwona Rotter

**Affiliations:** 1Chair and Department of Medical Rehabilitation and Clinical Physiotherapy, Pomeranian Medical University in Szczecin, Żołnierska 54, 70-210 Szczecin, Poland; 2Department of Biology and Medical Parasitology, Pomeranian Medical University in Szczecin, Powstańców Wielkopolskich 72, 70-111 Szczecin, Poland

**Keywords:** bone, cadmium, copper, iron, mercury, zinc

## Abstract

The paper presents the current understanding on the effects of five metals on bone tissue, namely iron, zinc, copper, cadmium, and mercury. Iron, zinc, and copper contribute significantly to human and animal metabolism when present in sufficient amounts, but their excess or shortage increases the risk of developing bone disorders. In contrast, cadmium and mercury serve no physiological purpose and their long-term accumulation damages the osteoarticular system. We discuss the methods of action and interactions between the discussed elements as well as the concentrations of each element in distinct bone structures.

## 1. Introduction

Bone is a highly mineralized tissue with considerable mechanical strength. Its intercellular substance contains inorganic components in the form of crystals, which sets it apart from other types of connective tissue. In addition to providing the environment for bone marrow, bone tissue also stores some substances, including essential elements, with significant biological functions. The creation of the organic components of bone laminae and their mineralization is carried out by osteoblasts, whereas osteoclasts facilitate bone remodeling through osteolysis [1,2].

Among the elements found in the environment, some metals, such as calcium (Ca), magnesium (Mg), phosphorus (P), fluoride (F), copper (Cu), zinc (Zn), and iron (Fe), are crucial for the body’s correct operation, while others, such as lead (Pb), cadmium (Cd), and mercury (Hg), have extremely harmful effects and obstruct the correct functioning of critical processes [3,4]. The amounts of various elements in the body are influenced by both biological and environmental factors, such as dietary consumption, gender, age, and environmental pollution. The effects of Ca, Mg, P, F, and Pb on the bone tissue have been already described in our previous paper [5], thus we focus on the influence of Fe, Zn, Cu, Cd, and Hg on the bone.

Fe, Zn, and Cu play an important role in human and animal metabolism. They participate in cellular metabolism, anti-inflammatory reactions, regulation of gene expression, protein synthesis, and mineralization and bone formation processes. Interactions between these metals affect their availability to the bone due to the similarity of their physicochemical properties. In contrast, Cd and Hg have no physiological function in the human body and their accumulation in human tissues and organs can adversely affect health. For example, both Cd and Hg have been found to contribute to bone mineral density loss [3,4,6,7]. This paper presents the latest data on the effects of Fe, Cu, Zn, Cd, and Hg on bone tissue, and discusses the correlations and mechanisms between these elements. 

## 2. Bone

### 2.1. Osteoblasts

Osteoblasts are bone-forming cells that are derived from multipotent mesenchymal stem cells (MSCs) [8]. The differentiation of MSCs into osteoblasts is a naturally occurring physiological process and is stimulated by multiple molecular signaling pathways, including the Wnt/β-catenin pathway, bone morphogenic proteins (BMPs), and pathways using mitogen-activated protein kinases (MAPKs) [9,10]. Recent studies have also highlighted the key role of reactive oxygen species (ROS) in osteogenesis as a common regulator of various osteogenic signaling pathways [11]. The above mechanisms lead to the activation of transcription of relevant genes. The focal point that receives biochemical signals and activates the MSCs differentiation into osteoblasts is the transcription factor Runx2. At the initial stage, MSCs differentiate into pre-osteoblasts, where Runx2 activates the transcription of genes encoding alkaline phosphatase (ALP), αI chain of type I collagen, and osteonectin [12]. Thus, pre-osteoblasts already show relatively strong expression of Runx2 and produce the basic components necessary for extracellular matrix (ECM) construction, including ALP, which is needed for the hydrolysis of organophosphates in order to utilize phosphate ions for the construction of bone mineral, and type I collagen, an essential protein for bone elasticity. In addition, Runx2 activates the transcription of another pro-osteogenic transcription factor, the Osterix protein. The Osterix factor enhances transcription of ECM proteins and is essential for the differentiation of osteoblasts from pre-osteoblasts and for skeletal mineralization in vivo [13]. Osterix transcriptional activity further differentiates pre-osteoblasts into immature osteoblasts, characterized by the production of osteopontin and bone sialoprotein. Subsequently, Runx2 expression decreases, allowing the differentiating cells to reach the mature osteoblast stage, characterized by osteocalcin (OCN) production [12,14].

### 2.2. Osteoclasts

Hematopoietic cells of the monocyte−macrophage line are the precursors of osteoclasts [15]. Osteoclastogenesis is a complex process consisting of the recruitment of precursor cells and their differentiation, fusion of preosteoclasts into giant multinucleated forms, and activation of immature osteoclasts [16]. In order for osteoclasts to form from precursor cells, the presence of two key cytokines is necessary: macrophage colony stimulating factor (M-CSF) and receptor activator of nuclear factor κB ligand (RANKL) [17]. At the first stage of osteoclastogenesis, M-CSF stimulates proliferation and prevents apoptosis of osteoclast precursors. PU.1 transcription factor and microphthalmia-associated transcription factor (MITF) play an important role at this stage [16]. PU.1 regulates gene transcription for both M-CSF and RANKL. In addition, by interacting with MITF, PU.1 also regulates the transcription of osteoclast-specific genes, such as the gene for cathepsin K, tartrate-resistant acid phosphatase (TRAP), and osteoclast-associated immunoglobulin-like receptor (OSCAR) [16]. RANKL is a key mediator in the other three stages: the differentiation, fusion, and activation of osteoclasts. RANKL is a ligand for the membrane receptor activator of nuclear factor κB (RANK) present on the surface of osteoclasts at the early stages of their differentiation. The combination of RANKL and RANK triggers a cascade of signals inside the nascent or mature osteoclast, necessary for its differentiation or resorptive activity [18,19]. RANKL/RANK signaling activates transcription factors such as nuclear factor κB (NF κB) and nuclear factor for activated T cells 1 (NFAT1). NFAT1 activation results in osteoclast differentiation [18]. TRAP, cathepsin K, matrix metalloproteinase 9 (MMP-9), and carbonic anhydrase II are also involved in osteoclast differentiation and bone resorption [15]. 

## 3. Essential Elements

### 3.1. Iron

Iron (Fe) is a micronutrient that is essential for the proper functioning of the body. Of the 3–5 g Fe found in an adult human body, 60–70% is bound to hemoglobin. Fe is also contained in ferritin and hemosiderin, proteins with a high affinity for Fe that are responsible for its storage in the liver, spleen, and bone marrow. The remainder is incorporated into myoglobin and many enzymes (catalases, peroxidases, and cytochromes) and proteins involved in transferrin transport. As a component of blood, Fe is involved in the transport of molecular oxygen from the lungs to all cells of the body. It also participates in erythropoiesis, leukocyte formation, and immune reactions, affecting the humoral and cellular immunity of the body [20,21,22]. The role of Fe in bone tissue is not well understood. It is known to be involved in collagen synthesis and the conversion of 25-hydroxyvitamin D to its active form. It is also essential for the proper functioning of osteoblasts, cells which are responsible for bone-forming processes [23,24]. 

The main source of Fe for humans and animals is food. The daily recommended Fe intake for humans depends on age, gender, physiological state, and physical activity; it ranges from 11 mg/day for infants to 19 mg/day for adults. The highest Fe concentrations are found in food products of animal origin, including liver, kidney, and bird egg yolks, and among plant products, Fe is most abundant in wheat bran [20,25].

Fe deficiency is generally a result of an inadequate supply of Fe, absorption disorders, or a consequence of certain diseases. The most characteristic symptom of Fe deficiency is anemia, a condition accompanied by fatigue, pallor, lack of appetite, thermoregulation problems, weakened immune system, and increased toxicity of metals with eminently poisonous properties, such as Cd. In vitro and in vivo experiments have shown that Fe deficiency disrupts bone homeostasis. It affects both bone formation and bone resorption, resulting in reduced bone mineral density and mass, altered microarchitecture, and reduced strength [26,27,28,29,30]. Studies by Angus et al. [31] and Michaelsson et al. [32] conducted on a population of pre- and postmenopausal women indicate a significant effect of Fe deficiency on bone mineral density (BMD). Toxqui et al. [33] report that Fe deficiency is associated with a higher rate of bone resorption. Animal studies have revealed that reduced dietary Fe supply causes unbalanced bone turnover, ultimately leading to weakened bones characterized by low BMD and reduced bone mineral content [27,29,34]. The exact mechanisms of iron deficiency on bone formation and resorption are not well understood. Katsumata et al. [27] report reduced serum concentrations of 1,25-dihydroxycholecalciferol, insulin-like growth factor I (IGF-1), and osteocalcin, which suggest that Fe deficiency reduces not only bone formation but also bone resorption, although bone resorption remains greater than bone formation, resulting in overall loss [27]. The latest data revealed that the effect of low Fe is biphasic; mild low Fe increases osteoblast activity, whereas very low levels of Fe inhibit osteoblast activity [35].

Genetic or acquired disorders of Fe metabolism, such as hemochromatosis, thalassemia, and sickle cell anemia, can lead to excessive Fe accumulation in the body. Fe overload is associated with bone weakness, which is represented as reduced bone mass, osteopenia, osteoporosis, altered bone microarchitecture and biomechanics, and frequent bone fractures [36,37]. A considerable percentage of patients with hemochromatosis suffer from osteoporosis (25–34%) and osteopenia (40–80%) [38]. 

Excess Fe affects both osteoblast and osteoclast metabolism and it has been better investigated than Fe deficiency. In osteoblasts, excess Fe exerts an antagonistic effect on preosteoblastic cells, disrupting cell-to-cell differentiation [39,40,41]. Recently, it has been shown that excess iron inhibits osteogenic differentiation of MSCs through the downregulation of Runx2 [22]. Fe excess reduces the expression of Runx2 and its downstream targets in human osteoblasts, namely OCN and ALP, ultimately leading to impaired extracellular matrix mineralization in osteoblasts [42]. Fe excess also decreases osteoblast activity by limiting the expression of the iron-modulated gene (HHIPL-2) [40,43,44,45]. It has been also found that osteoblasts respond to Fe overload by rapid and sustained down-regulation of the transferrin receptor and up-regulation of ferritin light and heavy chains (FtL and FtH). As a result, the osteoblast phenotype and function is suppressed [34]. Fe overload also results in the production of ROS which induce oxidative stress in osteoblasts leading to increased apoptosis [42,46,47]. 

In vivo studies have shown that Fe overload causes oxidative stress by increasing the expression of proosteoclastogenic cytokines such as tumor necrosis factor 𝛼 (TNF𝛼), interleukin-6 (IL-6), transforming growth factor 1 (TGF-1), and osteopontin (OPN), which stimulate osteoclasts through RANKL [48,49]. There is also evidence that Fe affects mature osteoclast activity and bone resorption. Mature osteoclasts exhibit high expression of TRAP, an Fe-containing enzyme marker of bone resorption [22,50]. TRAP-deficient mice have been found to have increased bone formation and elevated mineral density [51,52]. In vivo studies show that the bone loss observed in mice with Fe overload is due to accelerated bone destruction. An increase in the number of osteoclasts in bone tissue [53,54] and elevated levels of type I collagen C-telopeptide, a serum marker of bone resorption, have been found in hepcidin-deficient mice, suggesting that increased osteoclast activity induces bone loss due to Fe overload [22,55]. 

Zaichick and Zaichick [56] report the mean physiological bone Fe concentration in individuals from Central Europe at 55.5 mg/kg dw. However, most papers on Fe concentrations in different bone types indicate higher concentrations, ranging from 20.30 to even 276.33 mg/kg dw (Table 1). When comparing bone types, average Fe concentrations can be arranged in the following descending series: ribs > femur > tibia. Brodziak-Dopierała et al. [37] report higher Fe concentrations in the compact bone, while Zhang et al. [57] and Samudralwar [58] show higher Fe concentrations in the spongy bone. Some of the papers note greater Fe accumulation in the bones of male subjects, although most studies, for example by Budis et al. [59], Kot et al. [60], and Brodziak-Dopierała et al. [61], report no statistically significant differences between the genders. Probably, it is related also to the age of the studied patients. Differences in Fe bone content in men and women are less pronounced after the age of 45 with a decrease in estrogen production and lack of menstruation during menopause [62]. Some researchers have also observed differences in Fe levels between older and younger patients with the higher Fe content in older participants; however, these differences are usually small [56,63,64]. Among environmental factors, smoking, alcohol consumption, contact with chemicals at the workplace, place of residence, and having endoprosthesis were examined. Most studies did not reveal any impact of these factor on Fe concentrations in the bone tissue. However, some authors noted that smoking and alcohol consumption may affect bone Fe concentrations [59,60,63,65]. 

### 3.2. Zinc

Zinc is part of about 300 enzymes that are involved in the biochemical transformations of carbohydrates, proteins, and nucleic acids. There are 2 g to 4 g of this element in the human body, 60% of which is accumulated in the muscle, 20% in the bones, and 10–15% in the skin. Zn has a half-life of 162 to 500 days, and its level decreases with age [66]. Zn is responsible for the DNA-binding ability of many transcription factors through the unique ability to form molecules known as Zn finger proteins. Zn influences the function of many hormones, including testosterone, growth hormone, and gonadotropins. It plays a role in the synthesis, storage, and secretion of insulin. The metal is also a component of thymulin which is essential for T cell maturation and differentiation. Zn participates in the elimination of free peroxide radicals as a part of peroxide dismutase (Zn-SOD) and it inhibits the oxidation of unsaturated fatty acids [67]. As a cofactor, Zn is essential for the proper functioning and development of teeth as well as the skeletal system [66]. 

The recommended daily intake of Zn depends on several factors, including gender, age, sex, weight, and diet. Daily Zn supply according to WHO should be 6.5 mg/day for women and 9.4 mg/day for men (both over 19 years of age). Zn is best assimilated from meat products than from plant products. Food products containing the most Zn are calf liver, pork liver, pumpkin, and white beans. A high intake of animal proteins as well as the presence of lactose and highly processed products have a positive effect on Zn bioavailability. However, alcohol, folates, diuretics, alkaline drugs, Ca, Fe, and Cu have a negative impact on Zn absorption [67,68,69].

Zinc is unable to pass freely through the cell membrane, so it is transported via a Zn transporter (ZIP or ZnT) [70]. The latest data show an important role of Zn transporters in bone development, regeneration, and homeostasis. A total of 14 ZIP family members (ZIP1-ZIP14) and 10 ZnT family members (ZnT1-ZnT10) have been identified in mammalian genomes [71]. In vitro studies show that ZIP1 may play regulatory roles in stimulating osteogenesis and inhibiting osteoclastogenesis. ZIP1 is expressed in osteoblasts and later in ameloblasts and odontoblasts, indicating that ZIP1 is closely related to bone formation [72]. Additionally, ZIP1 is localized in the cytoplasmic membrane and mediates Zn influx during the differentiation of MSCs into osteoblasts [73]. However, ZIP1 is also expressed in osteoclasts, and overexpression of ZIP1 inhibits the activity of mature osteoclasts by negatively affecting the NF-κB pathway, which is essential for osteoclast differentiation and activity [74]. Mutations and disorders of ZIP and ZnT transport proteins may be associated with certain aspects of bone physiology or predisposition to certain bone diseases [75]. The zinc-ZIP8-MTF1 pathway is an important regulator of osteoarthritis pathogenesis [76] and dysfunctions of the *ZIP13*, *ZIP14*, or *ZnT5* genes cause bone deformity in both mice and humans [70]. 

Researchers agree that Zn is significant in the process of bone growth but the mechanism of action of this metal in the various stages of osteogenesis, as well as the effect of its deficiency on this process, are not well understood. 

Zn is important for bone growth and bone quality. Zn is a structural element of ossein, the organic extracellular matrix of bone. In addition, Zn is an important cofactor of the enzyme affecting the synthesis of various ossein components and plays a role in regulating bone resorption. Bones showing a lack of or extreme deficiency of Zn are thin, brittle, and characterized by increased resorption. Low Zn intake contributes to a gradual decrease in bone mass [77]. Rats administered a low Zn diet had reduced growth compared to the control group. Additionally, rats fed with low Zn diet had lower body weight and osteopenia [78,79]. Shortened long bones and reduced growth were observed in turkey fed a diet with reduced Zn concentrations. Additionally, reduced ALP activity was observed, and it was proportional to the level of Zn in the diet [80]. 

Zn also stimulates the synthesis and increases the effectiveness of growth hormone and IGF-1 on hard tissue. Some authors revealed that patients with short stature and dwarfism had Zn deficiency [81,82]. Many additional symptoms in Zn deficiency have also been described, such as abnormal rib and vertebral development, agenesis of long bones, club foot, cleft palate, micrognathia, ossification abnormalities, abnormal mineralization, and bending of long bones. Acrodermatitis enteropathica, a rare disease characterized by impaired Zn absorption, reveals the essential role of Zn in bone metabolism [83,84].

Zn has a strong osteoblast-stimulating and bone-forming effect and an inhibitory effect on bone resorption by osteoclasts. It can promote the differentiation and activity of osteoblasts and chondrocytes [85,86,87] and it activates aminoacyl-tRNA synthetase in osteoblast cells [88,89,90]. Zn affects the osteoblast proliferation and activity of ALP; however, the positive effects occur only in the narrow dose range (1–50 μM) [91]. Zn can also affect osteoblast differentiation by Zn finger transcription factors, Odd-skipped related 2 (Osr2) and Gli-similar 3 (Glis3) [92,93], and by modulating the expression of the *Runx2* gene [94,95]. It has been found that Zn induces *Runx2* through BMP signaling [96]. In contrast, Park et al. [97] suggest that Zn induces the expression of *Runx2* through a cAMP mediated process. There is also evidence that Zn protects osteoblasts from apoptosis. It was found that apoptosis rates in MC3T3-E1 cells increased from 7% in normal media, to 75% in zinc-deficient media, and to 90% in zinc-free media [98]. Apoptosis was induced by the mitochondrial (intrinsic) pathway through increased levels of cytochrome c [98]. Additionally, ZnT7 has been noted to be involved in the protection against apoptosis. The addition of Zn to culture media protected against H2O2-induced apoptosis by increasing expression of ZnT7 which leads to activation of protein kinase B (AKT) and extracellular-signal-regulated kinase (ERK) [99,100].

Zn is necessary for osteoclastogenesis. A 25% reduction of Zn content in bones was observed in rats fed a Zn-free diet. Additionally, 50% less osteoclasts were observed in the distal femur growth plate of the rats with Zn-free diet [101]. Osteoclastogenesis depends on the Zn concentration. The inhibition of osteoclastogenesis occurs at concentrations < 0.2 μM and >10 μM Zn [91]. Zn inhibits the process directly by reducing calcineurin phosphatase activity, which inhibits nuclear levels of dephosphorylated NFATc1, a critical factor for osteoclastogenesis, or by inhibiting NF-κB signaling [102]. Other authors [103,104,105] suggest that Zn inhibits the osteoclastogenesis indirectly by decreasing RANKL expression. 

Osteoclast differentiation and maturation depends on Zn levels. It has been found that bones of rats fed a Zn-free diet had reduced activity of cathepsin K and TRAP [101]. Reduced TRAP activity was also observed in in vitro and in vivo studies that examined the effects of excess Zn on osteoclasts. In contrast, Hadley et al. [106] noted increased activity of TRAP in rats administered an increased concentration of Zn in the feed. Authors also found decreased activity of MMP-2 and MMP-9, as well as carbonic anhydrase II, in which activity is essential for the resorptive action of osteoclasts [107,108]. Excessive levels of Zn can also induce apoptosis in osteoclasts. Li et al. [109] and Yamada et al. [110] observed increased osteoclast apoptosis when osteoclast were cultures with discs containing increased Zn concentrations. 

Trace element concentrations in human bones have been determined in a variety of bone types, with most work on the femur and rib [37,111,112,113,114,115]. However, the highest Zn values have been determined in the tibia (Table 1) [116]. Zaichick and Zaichick [56,115] present physiological levels of Zn in the femoral neck, iliac crest, and rib in Central European residents at, respectively, 55.5; 60.8, and 92.4 mg/kg dw. Roczniak et al. [116] also found a correlation between Zn concentration and type of the bone. 

Zaichick and Zaichick [114] report changes in Zn concentrations according to patients’ age, noting slightly higher concentrations of the metal in those under 35 years of age compared to those older than 55 (66, 1 and 65.7 mg/kg dw, respectively); however, most studies did not reveal the influence of patients’ age on Zn levels in the bone tissue. Similar data refer to gender-related differences. Zioła-Frankowska et al. [65] found higher Zn levels in male bones, while Zaichick [117] observed higher Zn levels in female bones. Other available data did not note any significant differences between women and men. Among biological factors, bone diseases and disorders were also examined. Milachowski et al. [118] found that patients with osteoporosis have lower Zn concentrations in their bones than patients with osteoarthritis, while Helliwell et al. [119] noted higher Zn levels in bone of patients with osteoporosis than in patients with fractures. Among environmental factors, smoking, alcohol consumption, contact with chemicals at the workplace, place of residence, and having endoprosthesis were examined. Kuo et al. [63] report an effect of various factors, including tobacco, alcohol, and seafood and fruit consumption, on the Zn concentration in the femoral head. Only Zioła-Frankowska et al. [65] confirmed the influence of smoking of Zn levels in the bone; they found higher Zn concentration in the femoral head of non-smokers.

### 3.3. Copper

The human body contains between 50 and 120 mg of copper, of which nearly 70% is found in muscle and skeleton [120,121]. The RDA for adult men and women is 900 μg/day [122], and the foods with the highest Cu content are bovine and sheep liver, dried fruit, cashew nuts, seafood, some fish (cod, mackerel), and chicken and turkey legs [123]. The minimum requirement for Cu is 0.4 to 0.8 mg/day [124]. Cu is involved in bone tissue metabolism, but its primary function is to participate in oxidation–reduction processes, where it acts as a coenzyme regulating Fe metabolism and transport and collagen metabolism [125,126,127]. It is a cofactor for many enzymes, including cytochrome c oxidase, and thus participates in the cellular respiration pathway. In mammals, Cu is also involved in collagen hardening, keratinization of hair and coat, and normalizes Ca and P deposition in bones [128]. Cu is involved in bone formation and mineralization, as it is also a cofactor of several enzymes, including lysyl oxidase that is responsible for the process of collagen fiber cross-linking, the disruption of which leads to bone weakness [37]. In vitro studies have shown the positive effects of Cu on cells that regulate bone metabolism. In addition, the presence of Cu stimulates the differentiation of MSCs toward the osteogenic lineage [129]. However, the dose of Cu is important. Li et al. [130] found that low Cu concentrations improved the viability and growth of osteoblastic cells, while higher Cu concentrations were found to be cytotoxic [131,132]. Li et al. [130] also showed that Cu^2+^ can inhibit osteoclast resorption.

Copper deficiency is extremely rare [133] and it leads to impaired melanin synthesis, impaired immune response, increased incidence of infections, cardiovascular disorders, and impaired cholesterol metabolism [128]. Symptoms of Cu deficiency also include reduced bone strength, impaired bone formation, and growth, reduced bone mineralization, reduced ossification of growth centers, and damaged cartilage integrity [134,135,136]. Animal studies have shown that Cu deficiency manifests as focal thickening of growth plates, interspersed with alternating non-calcified cartilage tissue and localized foci of increased ossification. Osteoporosis-like changes have been observed in the proximal part of the tibia in 10-week-old lambs born to ewes fed on Cu-deficient feed. Deficiency of this micronutrient also leads to a decrease in bone mass, as reflected by a thinning of the cortical layer of the bone visible on radiographic images. Destruction of bone structure makes bones brittle, resulting in reduced mechanical strength and fractures. Studies on Cu and Zn supplementation in humans and animals have led to the conclusion that deficiency of these micronutrients leads to osteoporosis-like changes. These elements increase BMD in menopausal women, which is why they are used in the therapy and treatment of osteoporosis [137,138,139]. 

Cu deficiency in bone metabolism is particularly pronounced in newborns affected by Menkes disease, a genetic disorder associated with severe Cu deficiency. This neurodegenerative disorder results in impaired Cu absorption, causing widespread effects, including bone lesions such as delayed growth, generalized osteoporosis, and exacerbated appendages of long bones [140]. The level of bone changes in Cu deficiency is mainly determined by functional defects in osteoblasts, while osteoclast activity remains unchanged [141,142]. Unaltered osteoclast activity in Cu deficiency, accompanied by low osteoblast activity, contributes to impaired bone tissue transformation leading to osteopenia [127]. The effects of Cu deficiency are mediated by various enzymes in which Cu is a cofactor, including amine oxidase, ceruloplasmin, cytochrome oxidase, dopamine monooxygenase, extracellular superoxide dismutase, lysyl oxidase, Cu/Zn superoxide dismutase, and tyrosinase. The main adverse effect attributed to Cu deficiency is the impaired activity of lysyl oxidase [134,135]. Decreased superoxide dismutase may also contribute to the inhibition of osteoblast activity, as it is sensitive to free radicals generated by osteoclast activity [84]. 

Cu, similar to Fe and Zn, plays many roles in the human body, but excess Cu can have serious negative effects. Excess Cu can generate ROS which induce lipid peroxidation and interfere with bone metabolism. The consequences include generalized loss of bone density, rickets, and anomalous osteophytes in patients with Wilson’s disease, a genetic disorder associated with impaired Cu metabolism [84]. It was found that patients with severe Wilson’s disease had lower serum ALP levels. However, Cu added in high concentration to serum in vitro did not have an important effect on serum ALP activity [143].

Excess Cu has an effect on osteoblasts. Moderate and high doses of CuCl_2_ 5H_2_O (100 μM and 150 μM, respectively) impair the osteoblast structure. Moreover, Cu inhibited the superoxide dismutase (SOD), glutathione peroxidase (GSH-Px) and ALP activity. Additionally, lower levels of collagen I, OCN, IGF-I, and BMP-2 in osteoblasts after additive Cu have been observed [144]. Qi et al. [144] also found that Cu promoted ROS production in osteoblasts. 

Excess Cu also has an influence on osteoclasts. In one study, a significantly reduced number of viable osteoclasts were detected at concentrations ≥ 20 µM Cu. It was found that Cu concentration is negatively correlated with number of osteoclasts. Additionally, activity of TRAP significantly increased in the presence of Cu, while the activities of cathepsin K and carbonic anhydrase II activity remained on the same level. Interestingly, Cu added to mature osteoclasts did not change the TRAP activity. The resorption was only moderately decreased [145]. 

Zaichick and Zaichick [56,114] established a reference Cu level in healthy people without chronic diseases from the central and non-industrialized parts of Russia. A value of 5–6 mg/kg dw was considered to be the physiological level of Cu in bone tissue, with the range varying according to people’s age and/or bone type. Table 1 shows the average Cu concentrations in the various bone types available in the scientific literature; Cu levels range from 0.16 to 6.30 mg/kg dw, with the highest value recorded in ribs. It is important to note that from the collected data presented in Table 1, 98% of the results are below the values considered physiological by Zaichick and Zaichick [56]. When comparing bone types, Cu concentrations can be arranged in the following descending series: tibia > femur > ribs (Table 1). Higher concentrations are recorded in the compact bone than in the spongy bone [37]. 

Cu concentrations in human bones have been shown to decrease with age [37,63]. Kuo et al. [63] noted age-related differences in Cu levels in the bones of the hip joint in Taiwanese patients (below 60 yrs—4.41 mg/kg dw, above 60 yrs—3.07 mg/kg dw, respectively). It has been found that Cu levels in patients with osteoarthritis are markedly lower than in healthy individuals [118], while Kuo et al. [63] did not reveal any differences in bone Cu level between bone disorders, e.g., femoral neck fractures, ischemic necrosis of the femoral head, osteoarthritis. However, the authors [63] found a correlation between Cu levels in the bones of the hip joint and the food consumption (internal organs, fruits, canned food). Brodziak-Dopierała et al. [37] found higher Cu levels in the femoral head of non-smokers than in smokers, and Zioła-Frankowska et al. [65] found higher Cu level in the femoral neck of patients who consumed alcohol. Place of residence, contact with chemicals, and gender had no effect on Cu bone content.

**Table 1 ijerph-20-02197-t001:** Iron, zinc, and copper levels in human bones in mg/kg dw (F, female; M, male; n, number of participants; A; alcohol consumers; NA, non-alcohol consumers; NS, non-smokers; S, smokers; O, osteoporosis; NO, no osteoporosis; * data converted from wet weight to dry weight assuming that the average water content of the bones is 30% of their weight).

The Study Area	Age	Sex	N	Fe Level	Zn Level	Cu level	Additional Information	Reference
Ribs								
Japan, Tokyo	61–96	F + M	45	71.00	149.00	0.19		[146]
		F	28	47.20	148.00	0.16		
		M	17	45.10	147.00	0.16		
Russia, Obninsk	15–55	F + M	80	140.00	92.80	1.35		[147]
	15–55	F	38	95.4	93.2	1.27		
	15–55	M	42	182	92.5	1.41		
Russia, Obninsk	15–55	F + M	84	276.33 *	189.67 *	2.07 *		[116]
Russia, Obninsk	15–55	F + M	80		92.80	1.05		[115]
		F	38		93.20	1.00		
		M	42		92.50	1.10		
Brazil	-	-	6		91.10			[49]
Ribs (spongy bone)								
USA, Kentucky	69 ± 6.3	F + M	12	77.00	144.00	1.40		[58]
Brazil, Sao Paulo	54.9	F + M	18		70.00			[148]
Ribs (cortical bone)								
USA, Kentucky	60–82	F + M	12	23.00	180.00	6.30		[58]
France	58–64	F + M	33		114.33	0.77	Control	[149]
					108.36	0.86	O	
Brazil, Sao Paulo	54.90	F + M	18		114.00			[148]
Sternum								
Poland, Katowice	26–55	F + M	35		92.10 *	0.47 *		[150]
Femur								
Poland, Silesia	67.5	F + M	50	139.77 *				[61]
	67.2	F	36	144.73 *				
	68.1	M	14	126.93 *				
Poland, Silesia	67.5	F + M	50		269.37 *	2.97 *		[116]
	67.2	F	36		273.47 *	2.90 *		
	68.1	M	14		258.73 *	3.10 *		
Femoral head								
Poland, Greater Poland Voivodeship	63.8	F + M	96	124.42	72.09	0.91		[65]
	64.5 ± 14.2	F	57	119.91	68.91	0.74		
	63.2 ± 10.2	M	39	131.01	76.75	1.16		
Poland, Upper Silesia	71 ± 6	F	64	118.36	85.10	4.22		[37]
		M	39	145.13	88.63	3.73		
Poland, Silesia, Lodz, Cracow	65.8 ± 12.5	F + M	197	67.30		0.81		[151]
Poland, Upper Silesia	71.6	F	69	118.36	85.10	4.22	Patients living in the industrial area	[113]
		M	39	145.13	88.63	3.73		
Taiwan	-	F + M	70	20.30	115.00	3.60		[63]
Great Britain, Liverpool	64–90	F + M	13		205.30		Control	[119]
	49–86	F + M	21		167.30		osteoarthritis	
	59–89	F + M	20		153.60		fracture	
Head of the femur (spongy bone)								
Poland, West Pomeranian Voivodeship	32–82	F + M	37		83.10	0.67		[7]
	32–82	F	24		85.30	0.72		
	53–78	M	13		79.10	0.58		
		F + M	5		78.20	0.62	O	
		F + M	32		83.90	0.68	NO	
Poland, West Pomeranian Voivodeship	32–82	F + M	37	49.80				[59]
Poland, Upper Silesia	71 ± 6	F + M	103	64.04	61.48	2.63		[37]
Poland, Upper Silesia	68 ± 9.9	F + M	13	50.27	83.63	0.66	None of the patients had ever been occupationally exposed to heavy metals	[62]
		F	9	41.88	82.83	0.58		
		M	4	67.04	85.23	0.81		
Poland, Lodz	68.3 ± 7.3	F + M	12	81.32	84.88	0.62	None of the patients had ever been occupationally exposed to heavy metals	[62]
		F	10	64.78	82.43	0.57		
		M	2	164.04	95.34	0.86		
Poland, Cracow	69.2 ± 9.6	F + M	13	47.89	101.10	0.58	None of the patients had ever been occupationally exposed to heavy metals	[62]
		F	10	53.87	91.86	0.59		
		M	3	27.94	131.88	0.55		
Poland, Lower Silesian Voivodeship	65.9 ± 10.8	F + M	21	77.73	83.59	0.79	NS	[152]
	62.8 ± 17.2		22	75.18	80.51	0.91	S	
Poland, Upper Silesia	67.3 ± 8.6	F	66		155.58			[153]
	61.4 ± 13.6	M	25		165.35			
China, Shanghai	62	F	1	77.20	95.00	0.87		[57]
Germany	-	F + M	200		106.86	1.52		[118]
Head of the femur (cortical bone)								
Poland, Upper Silesia	71 ± 6	F + M	103	93.89	94.72	3.73		[37]
Poland, Upper Silesia	67.3 ± 8.6	F	50		78.98			[153]
	61.4 ± 13.6	M	21		82.40			
China, Shanghai	62	F	1	72.40	101.00	1.57		[57]
Femoral neck								
Poland, Greater Poland Voivodeship	63.8	F + M	96	131.52	68.7	0.89		[65]
	64.5 ± 14.2	F	57	145.04	68.07	0.83		
	63.2 ± 10.2	M	39	111.75	69.63	0.97		
Russia, Obninsk	15–55	F + M	85	55.50	55.50			[56]
Turkey, Erciyes	73.9 ± 9.7	F + M	30	182.00	2.342		Fracture	[154]
	72.8 ± 6.0		30	108.00	3.145		osteoarthritis	
Tibial plateau								
Poland, West Pomeranian Voivodeship	65.75	F + M	33	58.03	98.90			[155]
	67	F	22	55.98	98.79			
	64.5	M	11	62.14	99.12			
Tibia								
Poland, Silesia	67.5	F + M	50	90.13 *				[61]
	67.2	F	36	96.82 *				
	68.1	M	14	72.93 *				
Poland, Silesia	67.5	F + M	50		292.87 *	2.33 *		[116]
	67.2	F	36		285.53 *	2.67 *		
	68.1	M	14		311.77 *	1.50 *		
Tibia (spongy bone)								
Poland, West Pomeranian Voivodeship		F + M	44	56.03				[60]
	73.1 ± 8.2	F	32	55.00				
	73.5 ± 8.3	M	12	58.77				
		F + M	7	34.36			S	
		F + M	37	60.13			NS	
		F + M	7	28.22			A	
		F + M	37	61.29			NA	

## 4. Heavy Metals

### 4.1. Cadmium

Cadmium is an element that poses a major threat to human and animal life and health. Due to its considerable use in industry (production of nickel–cadmium batteries, dyes, stabilizers, plastics, protective coatings, fireworks, and fluorescent paints), Cd is one of the most ubiquitous toxic elements. Its long half-life element results in its ongoing accumulation in the environment and increases the risk of absorption in high concentrations [156]. The two main routes of absorption of this element are the inhalation route (10–40% of the absorbed dose), with cigarette smokers at the highest risk of exposure (burning single cigarette results in 0.1–0.2 µg of Cd entering the body) and the oral route (about 6% of the absorbed dose)—mainly through the consumption of plants, fish meat, shellfish, and offal. According to WHO/FAO recommendations, the acceptable dose of Cd is 60–70 µg/day and the tolerable intake is 0.4–0.5 mg/week [157,158]. 

Cd exhibits nephro- and hepatotoxic properties leading to functional changes in organs. Its osteotoxic effects were first observed in Japan in the 1960s in 90% of elderly women (after menopause and multiple pregnancies) living in areas contaminated with Zn and Pb ore waste. The condition, known as itai-itai disease, was first described in the population of the floodplain of Japan’s Jinzu River, the consumers of rice grown in fields fertilized with silt from local industrial plants [159,160]. At that time, the concentration of Cd reached 8 mg/kg dw in the soil and 2.7 mg/kg in the rice (the average was 0.5 mg/kg dw). The patients suffered from vitamin D-resistant osteomalacia, accompanied by severe pain in the sacrum, lower extremities, and ribs, spontaneous fractures, as well as renal tubular dysfunctions—proteinuria, glycosuria, and reduced sodium resorption [159,160,161,162,163,164,165,166,167,168,169]. Cherry [170] showed that Cd concentrations in the bones of people with the itai-itai disease were twice as high as in those not chronically exposed to the metal. The results of epidemiological and clinical studies indicated a link between the disease and Cd exposure, while the Cooperative Research Committee on the itai-itai disease (1967) suggested that a diet low in protein and Ca and polygenism may have been additional factors [171].

In vivo studies on experimental animals have shown that Cd exposure reduces mineralization and bone density, altering its biomechanical properties, thus making it more susceptible to deformation and fracture [171,172,173,174,175,176,177]. Cd compounds administered to pregnant rats exert embryotoxic and teratogenic effects on their offspring. Birth defects usually appear in the head and limb region—the most commonly described are skull and vertebral deformities and the lack of metatarsal, tibia, and humerus bones [127,178,179]. Cd administered in drinking water to male laboratory rats (1 or 4 μg/kg body weight per day for six months) contributes to changes in bone metabolism and structure, characteristic of disrupted mineralization processes [180]. Experimental studies conducted by Kogan et al. [181], during which cadmium chloride and cadmium sulfate were administered subcutaneously to rats for 12 months (at daily doses of 1 μg/kg body weight), revealed the osteotoxic effects of Cd on bone in the form of osteoporosis-like changes and demineralization of bone tissue. Cd-exposed rats have reduced BMD, trabecular thickness, increased trabecular space, and increased osteoclast activity compared to non-exposed control [178]. Cadmium’s toxic effects are also manifested by bone deformities and growth disorders in animals [161,164,171,178,182,183]. 

Although Cd toxicity is well understood, its mechanisms of action remain unclear. It has been suggested that Cd acts directly on bone tissue [164] by acting directly on bone cells [184] and/or indirectly by inducing renal failure, leading to increased Ca and P excretion, thus reducing vitamin D synthesis [166,185,186]. Due to similar properties to Ca, Cd can interfere with Ca metabolism during osteogenesis and bone homeostasis, increasing calciuria and disrupting calciotropic hormones [183,187,188]. Accordingly, Ca deficiency has been found to increase Cd toxicity [83]. In addition to disrupting the calcium–phosphate balance, Cd reduces Fe, Cu, and Zn concentrations [189]. Cd has also been shown to reduce the expression of osteoblast differentiation markers (Runx2, osteocalcin), extracellular bone matrix proteins (type I collagen), and enzymes involved in the mineralization process (ALP) [183]. In addition, in vitro studies have shown that Cd induces osteoblast apoptosis by disrupting the cytoskeleton [190], as well as DNA fragmentation, increasing the number of micronuclei and bridge nuclei [191] and increasing the amount of ROS through the activation of the p38 MAPK pathway and inhibition of the Erk1/2 pathway [192,193]. Additionally, in in vitro studies, incubation of osteoblasts with Cd induced the nuclear factor erythroid 2-related factor (Nrf2) activation, presumably to increase transcription of antioxidant-responsive genes to combat oxidative stress but also impaired the secretion of type 1 procollagen, osteocalcin, and ALP [193]. 

Some studies provide evidence that chronic Cd exposure reduces bone volume and increases the percentage of TRAP [194,195]. Mice exposed to Cd show altered bone structure and an increased number of osteoclasts, indicating increased expression of osteoclast marker proteins in the bone marrow, including TRAP, MMP-9, cathepsin K, and carbonic anhydrase [196]. In addition, the increased number and activity of osteoclasts may be due to increased serum parathormone levels and increased expression of RANKL [83,197]. 

Cd accumulates differently depending on the type of bone [116]. Based on the data collected in Table 2, Cd concentrations can be arranged in the following descending series: ribs > femur > tibia. The difference between the highest concentration and the lowest concentrations recorded in bone tissue is significant, namely two orders of magnitude (Table 2). Research on Cd levels in human bone preparations shows that this element has a higher affinity for accumulation in the spongy bone than in the cortical bone [153,198]. When comparing heavily urbanized, industrial areas and unpolluted areas, Cd concentrations are an order of magnitude higher in urbanized areas. Cd levels are not significantly related to gender, age, BMI, fish consumption, diary product consumption, and osteoporosis [116,198,199]. Some authors pointed to the effect of smoking on higher Cd levels in the bone tissue [116,153,155]. 

### 4.2. Mercury

Mercury (Hg) occurs in a metallic form as well as inorganic and organic compounds. In the aquatic environment, metallic and inorganic Hg is transformed by biochemical transformations (methylation) in microorganisms (aerobic bacteria) into organic forms, mainly methylmercury. The toxicity of Hg depends on its chemical form; the most toxic forms are organic Hg compounds. Similar to Cd, Hg has a high affinity for the sulfhydryl groups of proteins. Hg compounds interfere with enzymatic and hormonal reactions in the human body. The element has been recognized as a potent neurotoxin that primarily disrupts the central peripheral nervous system [200]. Some attention has also been drawn to the deposition of Hg in bone and cartilage tissue [56,115,201] but the mechanism of this process is not fully understood. Most likely, Hg ions are incorporated in place of calcium ions into carbonates or hydroxyapatites. There is also a hypothesis about the presence of MeHg in the organic part of bone [7,202]. However, no toxic effects of MeHg on osteocytes have been reported, and their accumulation of Hg is lower than in other cells of the human body [111,203]. 

There are little data in the available scientific literature on the osteotoxic effects of Hg on mammalian bone tissue. In pregnant rats and mice subject to mercuric chloride inhalation, the offspring had bone lesions, reduced ossification [204], as well as microstructural changes in the alveolar bones [205]. Nunes [205] also suggested that in rats, mercuric chloride inhalation results in changes in organic and inorganic components in the alveolar bone. Abd El-Aziz et al. [206] show that prenatal poisoning of experimental animals with organic Hg negatively affects the development of the rat fetus, delaying ossification and reducing the length of long bones. Based on research on animals, it has been concluded that Hg exerts both direct and indirect effects on bone turnover, accelerates bone mineral density loss and osteoporosis or osteomalacia due to estrogen deficiency, and interferes with chondrocyte metabolism, resulting in reduced long bone development [207]. Yachiguchi et al. [208] examined the influence of Hg on the bone cells using the osteoblasts and osteoclast in the scale of the marine teleost as a model system of bone. They found increased level of metallothioneins (a metal-binding protein in osteoblasts) and decreased expression of TRAP and ALP.

There are no reports in the literature on the effects of Hg on bone tissue in humans. Rasmussen et al. [209] determined Hg concentrations in human bone samples from the medieval period; however, because the material was obtained from archaeological sites, it cannot be compared with Hg concentrations in samples obtained from people living today, with quite different diets, occupations, ancestry, environmental pollution, and treatment modalities (in medieval times Hg was used to treat diseases such as leprosy and syphilis) [202,209]. Rasmussen et al. [209] report higher Hg concentrations in the spongy bone than in the compact, which may be a result of the increased metabolism of the spongy bone and the increased area of contact with blood vessels through which Hg can be transported [210,211,212]. The lowest Hg concentrations in men aged 17–19 years and 38–45 years have been determined in the long bones of the shoulder and knee joints. The highest Hg concentrations were found in the hip joint and the bones that make up the pelvis since these elements consist of the spongy bone [209]. 

Data on Hg concentrations in human bone formations are scarce. Moreover, they very often indicate a result below the detection limit of the measuring device. The highest concentrations of Hg have been determined in the ribs of patients who lived for 10 years in areas near a hazardous waste incinerator [213]. When comparing the long bones, there was a slightly higher concentration of Hg in the tibia compared to the femur (Table 2). The evaluation of biological and environmental factors on bone Hg concentrations, showed no effect on the patient’s age, weight and BMI, osteoporosis, the number of cigarettes smoked, having dental amalgamators, and the consumption of fish and seafood [199,214]. 

**Table 2 ijerph-20-02197-t002:** Cadmium and mercury levels in human bones in mg/kg dw (F, female; M, male; HWI, hazardous waste incinerator; n, number of participants; NS, non-smokers; S, smokers; * data converted from wet weight to dry weight assuming that the average water content of bones is 30% of their weight).

The Study Area	Age	Sex	N	Cd Level	Hg Level	Additional Information	Reference
Ribs							
Japan, Tokyo	61–96	F + M	45	0.28			[146]
		F	28	0.19			
		M	17	0.13			
Russia, Obninsk	15–55	F + M	84	0.09 *			[117]
Russia, Obninsk	15–55	F + M	85		≤0.0048		[114]
Russia, Obninsk	15–55	F + M	80	0.04	≤0.018		[115]
		F	38	0.04	<0.01		
		M	42	0.04	≤0.018		
Spain, Tarragona		F + M	22	0.17 *	≤0.17 *	People who had lived for 10 years near HWI	[215]
Spain	~51	F + M	20	0.13 * (in 1998)	<0.17 * (in 1998)	People who had lived for 10 years near HWI	[213]
				0.17 * (in 2003)	<0.17 * (in 2003)		
				0.13 * (in 2007)	0.17 * (in 2007)		
				<0.08 * (in 2013)	<0.17 * (in 2013)		
Ribs (spongy bone)							
USA, Kentucky	69 ± 6.3	F + M	12	2.4			[58]
Ribs (cortical bone)							
USA, Kentucky	60–82	F + M	12	2.7			[58]
Femur							
Poland, Silesia	67.5	F + M	50	0.07 *			[116]
	67.2	F	36	0.03 *			
	68.1	M	14	0.07 *			
Poland, Silesia	55–78	F + M	17		0.008		[214]
Femoral head							
Poland, Upper Silesia	71 ± 6	F	64	0.56			[37]
		M	39	0.46			
Poland, Silesia, Lodz, Cracow	65.8 ± 12.5	F + M	197	0.07			[151]
Poland, Upper Silesia	71.6	F	69	0.56		Patients living in the industrial area	[112]
		M	39	0.46			
Poland, Greater Poland Voivodeship	20- > 80	F + M	95		0.017		[216]
	20- > 80	F	57		0.016		
	41- > 80	M	38		0.020		
Taiwan	-	F + M	70	1.20			[63]
Head of the femur (spongy bone)							
Poland, West Pomeranian Voivodeship	32–82	F + M	37	0.03	0.002		[7]
	32–82	F	24	0.03	0.002		
	53–78	M	13	0.02	0.002		
Poland, West Pomeranian Voivodeship	32–82	F + M	22	0.03		S	[199]
		F + M	15	0.02		NS	
		F + M	5	0.02	0.002	Osteoporosis	
		F + M	32	0.03	0.002	Non-osteoporosis	
Poland, West Pomeranian Voivodeship	32–82	F + M	30	0.04			[198]
	32–82	F	20	0.04			
	46–78	M	10	0.03			
Poland, Upper Silesia	71 ± 6	F + M	103	0.26			[37]
Poland, Upper Silesia	68 ± 9.9	F + M	13	0.05		None of the patients had ever been occupationally exposed to heavy metals	[62]
		F	9	0.06			
		M	4	0.05			
Poland, Lodz	68.3 ± 7.3	F + M	12	0.03		None of the patients had ever been occupationally exposed to heavy metals	[62]
		F	10	0.03			
		M	2	0.05			
Poland, Cracow	69.2 ± 9.6	F + M	13	0.06		None of the patients had ever been occupationally exposed to heavy metals	[62]
		F	10	0.07			
		M	3	0.05			
Poland, Lower Silesian Voivodeship	65.9 ± 10.8	F + M	21	0.057		NS	[152]
	62.8 ± 17.2		22	0.061		S	
Poland, Upper Silesia	67.3 ± 8.6	F	66	0.85			[153]
	61.4 ± 13.6	M	25	1.17			
Head of the femur (cortical bone)							
Poland, West Pomeranian Voivodeship	32–82	F + M	30	0.03			[198]
	32–82	F	20	0.03			
	46–78	M	10	0.04			
Poland, Upper Silesia	71 ± 6	F + M	103	0.36			[37]
Poland, Upper Silesia	67.3 ± 8.6	F	50	0.46			[153]
	61.4 ± 13.6	M	21	0.67			
Femoral neck							
Poland, Greater Poland Voivodeship	20-> 80	F + M	95		0.026		[216]
	20-> 80	F	57		0.025		
	41-> 80	M	38		0.031		
Russia, Obninsk	15–55	F + M	85		≤0.0063		[57]
Tibial plateau							
Poland, West Pomeranian Voivodeship	65.75	F + M	33	0.05	0.005		[155]
	67	F	22	0.04	0.005		
	64.5	M	11	0.06	0.0043		
		F + M	15	0.03	0.003	NS	
		F + M	18	0.06	0.01	S	
Poland, Silesia	67.5	F + M	50	0.07 *			[116]
	67.2	F	36	0.07 *			
	68.1	M	14	0.07 *			
Poland, Silesia	55–78	F + M	17		0.009		[214]

## 5. Interactions

Different elements co-occur in varying amounts in tissues and organs, often manifesting synergistic or antagonistic relationships, thus contributing to the disruption of their narrow optimal concentration range and resulting in secondary deficiency or toxicity [217]. There is a synergism between Cu and Fe, and there is antagonism between Cu and Zn as well as between Zn and Fe [218]. 

The Cu–Zn relationship explains many of the symptoms associated with Cu deficiency. Increased Zn and Cu excretion causes various metabolic disorders, including inappropriate lipid metabolism leading to coronary artery disease or psychiatric disorders [217,218]. Zn and Cu have the antagonistic action. Zn is used to treat Wilson’s disease (condition with excessive Cu accumulation in the liver) as Zn supplementation may reduce Cu concentration in the body [219,220,221]. Sandström et al. [222] found that high doses of Zn supplementation also normalized the Cu concentration in patients with acrodermatitis enteropathica patients. In the bones, only Roczniak et al. [116] found antagonistic interaction between Cu and Zn. Many more authors, including Lanocha et al. [7], Kuo et al. [63], and Brodziak-Dopierała [37], noted a synergism between Cu and Zn in the bone. 

Zn also interacts with Fe [128]. An antagonistic Fe–Zn interaction has been established due their competition for a specific transport protein in intestinal absorption. Zn deficiency may also cause Fe-deficiency anemia and Fe accumulation in tissues and cells [223,224,225]. Jurkiewicz et al. [62] found moderate negative correlation between Fe and Zn in the femoral head of patients from Cracow, Poland. They did not observe the Fe–Zn correlation in bone tissue of patients from Silesian region and Łodz (Poland) [62]. Similarly, Zioła-Frankowska et al. [65], Kuo et al. [63], and Brodziak-Dopierała [37] did not confirm any relationship between these two elements in the bone tissue.

The relationship between Fe and Cu has been recognized for many years. The best-known link is provided by ceruloplasmin, a major Cu carrying protein, that is essential for the Fe mobilization from storage tissues. Decreased Cu status, in some cases, can generate anemia because of decreased tissue Fe release. This kind of anemia is responsive to dietary supplementation with Cu but not Fe. It has been shown that dietary iron absorption and the intestinal Fe transport pathway also requires the presence Cu [226]. Positive correlations between Cu and Fe were also found in the bone tissue [63,65]. 

The scientific literature describes antagonistic interactions between Cd and biologically essential elements: Fe, Cu, and Zn, resulting in metabolic disorders. The interaction between Cd and Cu results in the development of anemia [227,228]. Additionally, it was found that Cd can decrease Cu absorption by tissues; at the same time, Cu is replaced by Cd in the active centers of enzymes [229]. The interaction between Cd and Cu or Zn is due to the high affinity of Cd for metallothioneins and the ability of Cu and Zn to induce the production of this protein. These micronutrients protect cells from the toxic effects of Cd by reducing the accumulation of its ions, which occurs due to antagonism of Cu and Zn ions in relation to Cd in cellular transport. The antagonistic effect of Cd against Zn results in impaired synthesis and release of digestive enzymes and insulin, whose de novo production requires the presence of Zn ions [230,231]. The mechanism of Cd–Zn antagonism mainly involves the displacement of Zn by Cd from protein compounds. However, in the bones, synergistic correlations between Cd and Zn [7,112,116] and between Cd and Cu were found [7,37,62,63]. Cd–Zn correlations were not confirmed by Kuo et al. [63], Jurkiewicz et al. [62], and Brodziak-Dopierała [37]. 

Cadmium also influences Fe metabolism. It was shown that Cd can decrease liver concentration of Fe [84]. Brodziak-Dopierała et al. [37] found a positive Fe-Cd interaction but only in the cortical bone. In the trabecular bone, this interaction was insignificant [37].

Interactions are mostly studied between essential and toxic metals. There are very little data on the correlation between the two heavy metals. Interestingly, Lanocha et al. [7] found a negative correlation between Cd and Hg in the spongy bone. 

The summary of the effects of Fe, Zn, Cu, Cd, and Hg on bone metabolism, including the interactions between elements in the bone is presented in Table 3.

## 6. Conclusions

The biochemistry of the skeleton is very complex and relies on balances between cells, organic molecules, and inorganic components. The elements found in the environment can be essential, harmful, or both, depending on the intake level. In some cases, an excess or deficiency affects the bone metabolism. Despite many publications in this area, much remains to be discovered to fully clarify the effects of minerals on bone metabolism. 

The presented review shows the influence of three essential elements (Fe, Cu, and Zn) and two heavy metals (Cd and Hg) on bone health. We also presented the concentrations of these five metals in various bone. We described whether biological and environmental factors have an impact on metal concentration in bone tissue. We took into consideration only those factors which were examined in publications, including, gender, age, BMI, bone disorders, dental amalgamators, endoprosthesis, place of residence, smoking habit, alcohol consumption, fish, and seafood consumption, and contact with chemicals in workplace. These are not all sources of elements, and we suggest that in future studies, other sources of presented metals should be included. Our research also shows interactions between Fe, Cu, Zn, Cd, and Hg, where the changes of one element’s levels may significantly affect the concentrations of the other.

The main objective is to characterize the mechanisms of elements on the bone cells in order to present possible directions for further research and constantly deepen knowledge in this area.

## Figures and Tables

**Table 3 ijerph-20-02197-t003:** Summary table of the effects of Fe, Zn, Cu, Cd, and Hg on bone metabolism, including the interactions between elements (+, synergism; -, antagonism; +/−, in some studies it was found as synergism, in others as antagonism).

Metal		Physiological Levels in the Bone Tissue	Mechanisms and the Influence on the Bone Tissue and Bone Cells		Interactions with Other Metals in the Bone
			Excess Amount	Deficiency	
Essential	Iron (Fe)	55.5 mg/kg dw	**Osteoblasts**: (1) ↓ Runx2, ALP, OCN, HHIPL-2, (2) ↑ ROS (3) ↑ apoptosis **Osteoclasts**: (1) ↑ TNF𝛼, IL-6, TGF-1 and OPN	**The bone tissue**: (1) ↓ serum 1,25-dihydroxycholecalciferon, IGF-1 and osteocalcin	Fe-Cu (+) Fe-Zn (−) Fe-Cd (+)
	Copper (Cu)	5–6 mg/kg dw	**Osteoblasts**: (1) ↓ SOD, GSH-Px (2) ↓ ALP (3) ↓ collagen I, OCN, IGF-1, BMP-2, TGF-β1 (4) ↑ROS **Osteoclasts**: (1) ↑ TRAP		Cu-Fe (+) Cu-Zn (+/−) Cu-Cd (+)
	Zinc (Zn)	**The femoral neck**: 55.5 mg/kg dw **The iliac crest**: 60.8 mg/kg dw **The rib**: 92.4 mg/kg dw	**Osteoclasts**: (1) ↓ osteoclastogenesis (2) ↓ resoption (3) ↓ carbonic anhydrase II (4) ↓ MMP-2 and MMP-9 (5) ↑ apoptosis	Osteoblasts: (1) ↓proliferation (2) ↓ALP (3) ↓Runx2 (4) ↑apoptosis **Osteoclasts**: (1) ↓ TRAP, cathepsin K	Zn-Cu (+/−) Zn-Fe (−) Zn-Cd (+/−)
Toxic	Cadmium (Cd)		**Osteoblasts**: (1) ↓ Runx2, OCN, extracellular bone matrix proteins, ALP; (2) ↓ Nrf2; (3) ↑apoptosis **Osteoclasts:** (1) ↑ MMP-9, TRAP, cathepsin K, and carbonic anhydrase		Cd-Fe (+) Cd-Cu (+) Cd-Zn (+/−) Cd-Hg (−)
	Mercury (Hg)		**Osteoblasts**: (1) ↓ ALP (2) ↑ mettalothionenin **Osteoclasts:** (1) ↑ TRAP		Hg-Cd (−)

## Data Availability

Not applicable.

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
