# Peer review of "Iron, Zinc, Copper, Cadmium, Mercury, and Bone Tissue"

_ijerph, 2023, doi:10.3390/ijerph20032197_

Round 1

Reviewer 1 Report

The narrative review article by Ciosek, Kot, and Rotter explores the relationship between heavy metals and bone metabolism. The topic of the review is interesting. However, the article is unfocused and is extremely difficult to follow. A more focused analyses on bone, and bone resident cells (osteoclast, osteoblasts, osteocytes, MSCs) would strengthen the review. The authors should consider creating a table that outlines the studies focused on each metal on bone metabolism. This would be more helpful than the tables listing the metal concentrations in bone. Finally, the authors make generalized statements that need to be more precise. For example, line 80-81 says reduced regulatory factors and bone makers. The regulatory factors and bone markers need to be included. There are many examples of this throughout the text. Ultimately, this review will require significant revisions before it is suitable for publication. 

Author Response

Thank you for your review of our paper. We have answered each of your point below.

(1) The narrative review article by Ciosek, Kot, and Rotter explores the relationship between heavy metals and bone metabolism. The topic of the review is interesting. However, the article is unfocused and is extremely difficult to follow. A more focused analyses on bone, and bone resident cells (osteoclast, osteoblasts, osteocytes, MSCs) would strengthen the review. We changed the text and we do hope that now it is easier to follow.

(2) The authors should consider creating a table that outlines the studies focused on each metal on bone metabolism. This would be more helpful than the tables listing the metal concentrations in bone. We added some figures to present the effects of elements on bone cells. We did not create table (as a reviewer suggested) because Rodríguez and Mandalunis (J Toxicol 2018; 4854152) presented the effects of metals on bone tissue in a table, and we did not want to copy it. 

(3) Finally, the authors make generalized statements that need to be more precise. For example, line 80-81 says reduced regulatory factors and bone makers. The regulatory factors and bone markers need to be included. There are many examples of this throughout the text. Ultimately, this review will require significant revisions before it is suitable for publication.  We changed sthe text and we hope it is now suitable for publication.

Reviewer 2 Report

This review, entitled "Iron, zinc, copper, cadmium, mercury and the bone tissue" summarizes the current understanding of the effects on bone of three metals involved in metabolism, iron, zinc and copper and also of two without any physiological role, cadmium and mercury.

Iron, zinc, and copper contribute significantly to human and animal metabolism when present in sufficient amounts, but their excess or shortage increases the risk of developing bone disorders. In contrast, cadmium and mercury serve no physiological purpose and their long-term accumulation damages the osteoarticular system. We discuss the methods of action and interactions between the discussed elements as well as the concentrations of each element in distinct bone structures.

Overall, this review manuscript has important repercussions since it emphasizes a very important aspect, the fact that the long-term accumulation of elements like cadmium and mercury damages the osteoarticular system whereas iron, zinc, and copper are needed for human metabolism, but their excess also increases the risk of developing bone disorders. The manuscript contains sufficient noteworthy information to justify its publication. I have only one minor point to improve the manuscript:

I think that the review should include a section about gender-specific differences and the effect on human reproduction.

Likewise, the conclusions do not emphasize clearly the main findings of the research and could be rewritten.

Author Response

Thank you for your review of our paper. We have answered each of your point below.

This review, entitled "Iron, zinc, copper, cadmium, mercury and the bone tissue" summarizes the current understanding of the effects on bone of three metals involved in metabolism, iron, zinc and copper and also of two without any physiological role, cadmium and mercury. Iron, zinc, and copper contribute significantly to human and animal metabolism when present in sufficient amounts, but their excess or shortage increases the risk of developing bone disorders. In contrast, cadmium and mercury serve no physiological purpose and their long-term accumulation damages the osteoarticular system. We discuss the methods of action and interactions between the discussed elements as well as the concentrations of each element in distinct bone structures. Overall, this review manuscript has important repercussions since it emphasizes a very important aspect, the fact that the long-term accumulation of elements like cadmium and mercury damages the osteoarticular system whereas iron, zinc, and copper are needed for human metabolism, but their excess also increases the risk of developing bone disorders. The manuscript contains sufficient noteworthy information to justify its publication. Thank you very much.

I have only one minor point to improve the manuscript: I think that the review should include a section about gender-specific differences and the effect on human reproduction. We add gender-specific differences if they were. We decided not to add information about the effect of metals on human reproduction because we wanted to focused only about the bone tissue.

Likewise, the conclusions do not emphasize clearly the main findings of the research and could be rewritten. We completely changed the conclusion section.

Reviewer 3 Report

Why did this manuscript focus on just Cu, Zn, Fe, Cd, and Hg?  There are many more minerals that positively affect or negatively impact bone health and mineralization, depending on the dosage, including Cu, Zn, Fe, Cd and Hg.  Thus, why did this manuscript choose just these 5 minerals to focus on and not others?  The reasoning must be established in the Introduction to justify the review narrowly focused on these particularly elements.  Readers will ask these questions about this manuscript.

The review mentions a non-exhaustive list of sources for Cu, Zn, and Fe, but very limited sources of Cd and Hg, besides cigarette smoking, urbanized areas for Cd, and fish and seafood for Hg.  But more sources of Cu, Zn, and Fe can be given to show how to get adequate amounts and/or avoid excessive amounts, along with more specific sources of Cd and Hg, in order to avoid them.

The Interactions section only discusses the interaction of Cu, Zn, Fe, and Cd, with each other, but neglect to discuss their interactions with other nutrients that affect bone health, such as Ca.

Review and research manuscripts typically end with a lengthy Discussion and sometimes also a short Conclusion, but a Discussion section was not included in this manuscript.

Author Response

Thank you for your review of our paper. We have answered each of your point below.

(1) Why did this manuscript focus on just Cu, Zn, Fe, Cd, and Hg?  There are many more minerals that positively affect or negatively impact bone health and mineralization, depending on the dosage, including Cu, Zn, Fe, Cd and Hg.  Thus, why did this manuscript choose just these 5 minerals to focus on and not others?  The reasoning must be established in the Introduction to justify the review narrowly focused on these particularly elements.  Readers will ask these questions about this manuscript. We have already published the review paper about the influence of Ca, Mg, P, F, and Pb on the bone tissue (Biomolecues 2021, 11 (4), 506). This paper is a second part, in which we focus on the different metals, which also have impact on bone tissue.

(2) The review mentions a non-exhaustive list of sources for Cu, Zn, and Fe, but very limited sources of Cd and Hg, besides cigarette smoking, urbanized areas for Cd, and fish and seafood for Hg.  But more sources of Cu, Zn, and Fe can be given to show how to get adequate amounts and/or avoid excessive amounts, along with more specific sources of Cd and Hg, in order to avoid them. We agreed that we did not include all sources of elements. However, we described only those factors which were already examined in available publications, including age, sex, place of residence, contact with chemicals in the workplace, smoking, alcohol consumption, endoprothesis, BMI. Our main goal was to present actual knowledge and which factors exactly have been already examined. We added information in the conclusion section that not sources of elements were discussed. 

(3) The Interactions section only discusses the interaction of Cu, Zn, Fe, and Cd, with each other, but neglect to discuss their interactions with other nutrients that affect bone health, such as Ca. It was the intention of the entire work to focus only on these five described elements. We did not want to confuse the reader by mentioning an element without describing it.

(4) Review and research manuscripts typically end with a lengthy Discussion and sometimes also a short Conclusion, but a Discussion section was not included in this manuscript. This is not meta-analysis. The aim of the study was only to collect data from the available literature and present them in one manuscript. We leave the discussion of the results to the authors who will use this work.

Round 2

Reviewer 1 Report

The article is still difficult to follow, often jumping from osteoblasts to osteoclast to osteoblasts when describing the effects of these metals on differentiation and activity. This could be significantly improved by separating out the sections to focus on an individual cell type in a single paragraph. The authors also state that the Rodríguez and Mandalunis reference includes a summary table, but it does not. The added figures to this manuscript do not much to the manuscript. 

Author Response

Thank you for your review of our paper. We have answered each of your point below.

The article is still difficult to follow, often jumping from osteoblasts to osteoclast to osteoblasts when describing the effects of these metals on differentiation and activity. This could be significantly improved by separating out the sections to focus on an individual cell type in a single paragraph. We separated the section and we describe the effect of metals on the osteoblasts in one paragraph and the effect of metals on the osteoclasts in the other one. Additionally, we add one unit about osteoblasts and osteoclast to give the readers information about mechanisms in osteoblastogenesis and osteoclastogenesis.

The authors also state that the Rodríguez and Mandalunis reference includes a summary table, but it does not. The added figures to this manuscript do not much to the manuscript. We want to apologize. The table is published by Dermience et al. (J Trace Elem Med Biol 2015, 32, 86-106). Nevertheless, we decided to add table (Table no. 3) instead of the figures.

We hope the revised version is now suitable for publication.

Reviewer 3 Report

In your response, if this is the second part, why is the first part that was published in Biomolecules (2021) not cited in the Introduction?  Please also note that meta-analyses are not the only articles that have a Discussion section, but nearly all research and review articles have a Discussion section.

Author Response

Thank you for your review of our paper. We have answered each of your point below.

As the authors mentioned that this manuscript was part 2 and part 1 wasan article published in Biomolecules, that article could be cited in theIntroduction.  (It does not necessarily need to be cited, but it doesadd helpful context.) We added sentence in the introduction section “The effects of Ca, Mg, P, F, and Pb on the bone tissue have been already described in our previous paper [5], thus we focus on the influence of Fe, Zn, Cu, Cd, and Hg on the bone.”

But since that article in Biomolecules did nothave a Discussion section, then this manuscript submitted to IJERPH does not necessarily need one.  We did not add the discussion section

Based on the other reviewer's comments, it can be accepted for publication.  Thank you.